# Avoiding Negative Side-Effects And Promoting Safe Exploration With Imaginative Planning

## Abstract

With the recent proliferation of the usage of reinforcement learning (RL) agents for solving real-world tasks, safety emerges as a necessary ingredient for their successful application. In this paper, we focus on ensuring the safety of the agent while making sure that the agent does not cause any unnecessary disruptions to its environment. The current approaches to this problem, such as manually constraining the agent or adding a safety penalty to the reward function, can introduce bad incentives. In complex domains, these approaches are simply intractable, as they require knowing apriori all the possible unsafe scenarios an agent could encounter. We propose a model-based approach to safety that allows the agent to look into the future and be aware of the future consequences of its actions. We learn the transition dynamics of the environment and generate a directed graph called the imaginative module. This graph encapsulates all possible trajectories that can be followed by the agent, allowing the agent to efficiently traverse through the imagined environment without ever taking any action in reality. A baseline state, which can either represent a safe or an unsafe state (based on whichever is easier to define) is taken as a human input, and the imaginative module is used to predict whether the current actions of the agent can cause it to end up in dangerous states in the future. Our imaginative module can be seen as a "plug-and-play" approach to ensuring safety, as it is compatible with any existing RL algorithm and any task with discrete action space. Our method induces the agent to act safely while learning to solve the task. We experimentally validate our proposal on two gridworld environments and a self-driving car simulator, demonstrating that our approach to safety visits unsafe states significantly less frequently than a baseline.

## 1 Introduction

Commonly, reinforcement learning (RL) (Sutton et al. (1998)) methods involve learning to solve specifically defined tasks by making the agent take a sequence of actions which maximize the long term rewards given by the environment. However, a crucial and a commonly ignored part of these methods is ensuring the safety of the agent and its surrounding. As RL research advances and starts being used for real-world tasks, having an exploitative policy whose main objective is to maximize the rewards over all else is likely to fail in assuring general safety and may lead to serious, irreparable damage.

The past few years have seen a rise in awareness and research on *AI safety* (Amodei et al. (2016)). Instead of focusing on extreme scenarios involving super-intelligent agents with unspecified objectives, this area of safety research focuses on practical issues with modern machine learning techniques. Specifically, within RL, AI safety focuses on training an RL agent to learn how to solve the intended task without causing any harmful or unexpected results. Our work proposes one method to make progress on two important problems in AI safety. The first problem we tackle is that of *safe exploration* (Moldovan & Abbeel (2012); Koller et al. (2018); Altman (1999)). Safe exploration involves the agent respecting the implicit safety constraints, not only at convergence but also throughout the learning process. Ensuring such safety constraints is critical for applying RL to physical robots, where poorly-chosen actions may break the robot or cause it to get stuck in certain states. Safe exploration requires an agent to predict which actions are dangerous and explore ways in which it

can avoid them. The second problem we focus upon is to make the agent act in accordance with the designer's intention, even when this intention is poorly defined. In many real-world scenarios, the ideal specification and details of the task are hard to articulate and specify. These nuanced details are hard to embed in a formal objective. In many situations, a reward-maximizing agent will ignore the intentions of the designer and take actions that exploit mistakes in the reward function which can potentially cause harm. As described in Clark & Amodei (2016), a poorly designed reward function can lead to the problem of *reward hacking* and *negative side effects* (Amodei et al. (2016); Krakovna et al. (2018); Turner et al. (2019)). In reward hacking, the agent learns to exploit and maximize the reward by finding an unexpected and unintended clever solution which does not match with the intentions of the reward designer. We focus on the problem of negative side effects, in which the designer mistakenly ignores potentially large factors of the environment which can be irreversibly damaged if altered by the agent. Explicitly penalizing every possible side-effect is highly impractical and labor-intensive. Not only does this approach lead to an extremely complex reward function, and even if such a safety-aware reward function could be designed, it would impinge upon the agent's ability to effectively solve the task. In order to avoid excessive human input and to avoid side effects in complex environments, we must develop a more practical and generalized approach.

Not all side-effects are created equal, and many are benign. Hence, we want to avoid only those side-effects which are irreversible or difficult to reverse. In real-world tasks, irreversible changes often correspond to serious damage. For example, imagine a robot tasked with cleaning a table with a glass vase on top. The cleaning robot could either (1) move the vase to a different region of the table and carry out the cleaning or (2) drop the vase on the floor causing permanent damage and continue cleaning. The side-effect from the first strategy, moving the vase, is safe, as it can be undone. In contrast, the second strategy causes an unsafe side effect, as the robot is unable to "unbreak" the vase. The key insight of this paper is that humans can imagine the consequence of their actions on their environment (Tolman (1948); Pfeiffer & Foster (2013)). Humans can then explicitly choose the ones which do not cause any harm. The act of classifying the side-effect as safe or unsafe is based on the prior experience of the person. For example, a child learns that dropping a glass vase is harmful by being warned by an adult, seeing a vase drop, or dropping a vase themselves. Hence, whenever a human tries to clean the table, they can imagine the outcome of dropping the vase and chooses to avoid taking that action.

In this paper, we propose to use of imaginative planning to prevent the agent from taking unsafe actions which cause negative side-effects. We model the transitions between states and use this transition function to create a roll-out graph which considers all possibilities that can occur as the agent progresses. We call this the *imaginative module* of the agent. We assume that a human initially specifies a small number of baseline safe states or unsafe states, which we will use in conjunction with the imaginative module. If we define the baseline state as a safe sate, an action $a_t$ is considered to be a "safe action" if there is at least one path from the next state (which it reaches after taking $a_t$) to the baseline state in the imaginative module. By this definition, if the action $a_t$ was an unsafe action then taking it would cause permanent change to the environment, hence not allowing it to reach the baseline state in the graph. If the baseline state is defined as an unsafe state, an action $a_t$ is considered to be a "safe action" if there are no paths from the next state to the specified baseline state in the imaginative module. By this, the action $a_t$ would be unsafe if taking it will cause an agent to always end up in the baseline state, More precisely, we say that $a_t$ is unsafe if all paths in the imaginative module from the next state lead to an unsafe state.

Every-time the agent takes an action, we check whether the action would be safe or not based on the definition. If the action is not safe, we act greedily among the safe action based on the value function. Our formulation can be used with any on/off policy algorithm with discrete actions. Moreover, in addition to providing safety throughout the learning process, our imagination module can also accelerate training, as it eliminates a certain unsafe portion of the state, action pairs.

We run experiments to confirm that our proposed method improves safety throughout learning and at convergence. We experiment our method on grid-world environments specifically designed for safety-related problem proposed by Leike et al. (2017) and on an autonomous car simulator part of the OpenAI gym (Brockman et al. (2016)). We experiment and test our imagination graph by incorporating it with an "unsafe" pre-trained policy and while training a policy using standard on-policy methods.

## 2 RELATED WORK

Due to their low sampling complexity, model-based reinforcement learning methods have been a population approach to solving complex tasks. Learning the dynamics model is sample efficient as they are trained using standard supervised learning techniques, allowing the use of off-policy data. However, policy optimization heavily relies on the accuracy of the learned dynamics, and is likely to overfit if the model is not precise. There has been a lot of previous work (Weber et al. (2017); Clavera et al. (2018); Kurutach et al. (2018); Hafner et al. (2018)) on trying to reduce the model bias, combining elements of model-based and model-free reinforcement learning and scaling the model using other approaches.

Our work explicitly focuses on using a model-based approach to tackle few important problems in AI safety. The field of AI safety (Amodei et al. (2016)) studies how AI agents can be modeled to avoid intended or unintended harmful accidents. An accident can be described as a situation in which the agent designed and deployed for the task doesn't adhere to what the human designer had in mind and produces harmful and unexpected result. Among various research problems listed by Amodei et al. (2016) which constitute the field of AI safety, we specifically focus on the problem of safe exploration (Moldovan & Abbeel (2012); Koller et al. (2018); Altman (1999)) and avoiding negative side effects (Amodei et al. (2016); Krakovna et al. (2018); Turner et al. (2019)).

The problem of safe exploration involves agent to respect the safety constraints, both while operation and learning. It involves agents to predict which actions will lead to dangerous states in the future, so as to safely explore the state space. Badly chosen actions may destroy the agent or trap it in states it can't get out of. Safe exploration is one of the most studied problem within the field of AI safety. There are various methods which involve preserving the reach-ability of the starting state (Moldovan & Abbeel (2012); Eysenbach et al. (2017)) or safe regions (Gillula & Tomlin (2012); Fisac et al. (2018)) to promote safe exploration. Other methods (Achiam et al. (2017); Chow et al. (2018); Dalal et al. (2018)) involve changing the optimizing criteria for policy optimization from expected rewards by introducing implicit constraints. These are based on a well studied formulation of constrained Markov Decision Process (CMDP) (Altman (1999)) in which the agents must satisfy constraints on expectations of auxiliary costs. However, these method require an explicit and clever design of the constraints, which can be difficult.

The importance of addressing the problem of negative side effect has been gaining a growing interest. The problem occurs when the designer mistakenly ignores potentially large factors of the environment which can be irreversibly damaged if altered by the agent while performing the main task. Explicitly penalizing every possible negative side-effect is intractable and can lead to extremely complex reward function design. An approach to solving this problem involves human oversight using inverse reinforcement learning (Ziebart et al. (2008); Hadfield-Menell et al. (2016)), demonstrations (Abbeel & Ng (2004); Hester et al. (2018)), or human feedback (Christiano et al. (2017); Warnell et al. (2018)). However, these methods are difficult to quantify and depend a lot on the diversity of settings in which they are performed. A common and more tractable approach is that of using empowerment (Klyubin et al. (2005); Gregor et al. (2016)) the agent has over the environment. This measures the control the agent has over it's environment which can be derived by calculating the mutual information between the agent's current action and the future states. To avoid irreversible side effects, the agent can be penalized for change in empowerment wrt. to a baseline state which we consider safe. Krakovna et al. (2018); Turner et al. (2019) propose an important methodology of penalizing deviation by proposing the usage of different baseline states and deviation measures.

## 3 BACKGROUND

We consider an agent making decision sequentially over an environment $E$ over discrete time steps $t = 1, ..., \mathcal{T}$. We model the decision making process using a Markov Decision Process (MDP), in which the agent experiences a state $s_t \in \mathcal{S}$, where $\mathcal{S}$ is a discrete set of states and takes an action $a_t$ at each time-step $t$. The action $a_t$ is decided by a stochastic policy $\pi(a_t|s_t)$ over a discrete set of actions $\mathcal{A}$. After taking each action, the agent receives a reward $r_t = \mathcal{R}(s_t, a_t)$ from the environment, and transitions to a new state $s_{t+1} \sim \mathcal{P}(s_t, a_t)$. We do not assume the agent to have access to the model ($\mathcal{R}(s_t, a_t)$ & $\mathcal{P}(s_t, a_t)$), and explicitly learn the model.

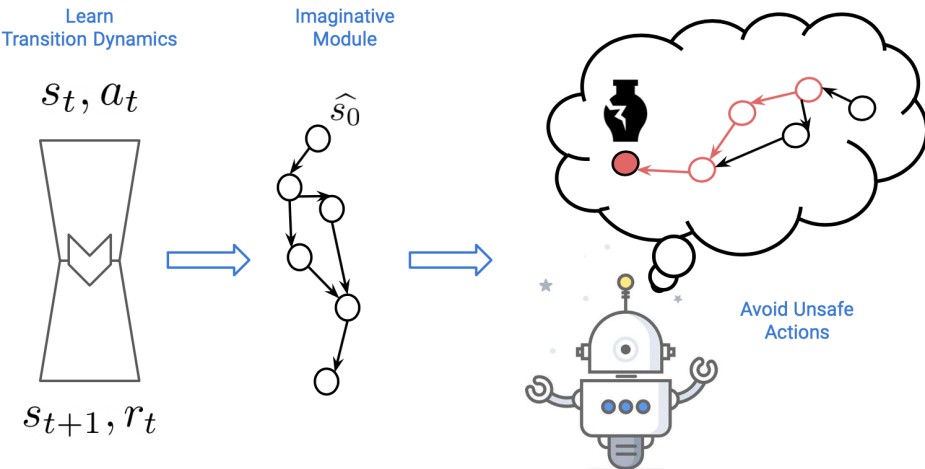

Figure 1: The proposed approach, illustrated. The transition dynamics of the environment is learned and a directed graph called imaginative module is created from the rollouts. This is then used to plan and check if the agent's next action will lead to negative consequences in the future.

## 4 IMAGINATIVE PLANNING FOR SAFETY

Humans utilize multiple learning, memory, and decision making systems in order to efficiently carry out the task. When the information of the environment is available or essentially required, it is found that the best strategy is model-based planning associated with prefrontal cortex (Daw et al. (2005)). When expecting implicit safety from the agent, it seems apparent that the information about the environment will be needed for the agent to know the impact of their actions. We plan to use this insight to make a deep reinforcement learning based agent aware of the implications of every action the agent takes and try to improve agent's decision-making capabilities in terms of safety and avoiding side effects.

In this paper, we consider making the agent initially learn about the environment and use this knowledge to check whether its actions would lead to negative permanent consequences. We initially make the agent learn the transition dynamics $s_{t+1} \sim \mathcal{P}(s_t, a_t)$ and the reward function $r_t = \mathcal{R}(s_t, a_t)$. We model these based on the recent successes of action-conditional next-step predictors (Hafner et al. (2018); Kaiser et al. (2019)), which take as input the current state (or history of observations) and current action, and predict the next state, and the next reward. These can be hence used to query about the future for efficient and safe decision making. Given the current state $s_t$ and the possible action $a_t$, we try to predict the next state $s_{t+1}$ and the reward $r_t$ that the agent will get. To try to cover all possible state-action pairs, we deploy a number of agents to explore different aspects of the environment randomly and collect the training data. Before testing/training the main policy, we generate a directed graph of all possible trajectories that can be followed by the agent using the transition dynamics we learned. The graph is generated by rolling out the transition model multiple time-steps into the future, by initializing the path with the initial state $s_o$ and subsequently feeding simulated states into the model. Each node of the graph consists of the imaginative state $\widehat{s_t}$ and the reward $\widehat{r_t}$, which the agent would get if it had visited that state. The edge set represents the actions that the agent can take, and for every node, we consider all possible discrete actions and use the learned transition model to query the next state $\widehat{s_{t+1}}$. We refer to this graph as the *imaginative module* of the agent. Querying multiple steps into the future involves the agent to search in the imaginative module and follow the respective path. Once the imaginative module has been created, we use it to explicitly tell the policy being learned/tested whether the action it's about to take is safe or not.

We define a baseline state(s) $\mathcal{B}$, which we will incorporate into planning done by the imaginative module to ensure safety. The baseline states are taken as a human input and based on the environment and the task, the baseline state can either represent a safe state or an unsafe state. For example, for the cleaning robot, the baseline state, which represents the safe state, could be the vase kept on the table (the initial state of the environment). For an autonomous car which has to learn how to drive without going off track, the baseline state represents an unsafe state of the car being out of track. How it

reaches outside the track is not required. If we define the baseline state as a safe sate, an action $a_t$ is considered to be a "safe action" if there's at least one path from the next state (which it reaches after taking $a_t$) to the baseline state in the imaginative module. By this definition, if the action $a_t$ was an unsafe action then taking it would cause permanent change to the environment, hence not allowing it to reach the baseline state in the graph. If the baseline state is defined as an unsafe state, an action $a_t$ is considered to be a "safe action" if there are no paths from the next state to the specified baseline state in the imaginative module. By this, the action $a_t$ would be unsafe if taking it will cause an agent to always end up in the baseline state. More precisely, we say an action is unsafe if all the paths from the next state lead to an unsafe state, as predicted by the imaginative module.

The main policy $\pi(a_t \mid s_t)$ which we learn/test is modeled using a convolutional neural network. This can either be a policy which we want to learn while ensuring explicit safety even during training or a pretrained unsafe policy to which we want to provide the ability to make safe decisions. To make the agent take a safe action, we initially predict the action it would take using the modeled policy $\pi(a_t \mid s_t)$. Then we plug in our imaginative module to determine whether the action predicted by the policy is safe or not. If the action is safe, the agent takes it. Otherwise, if the action is unsafe, the agent is made to act greedily with respect to the safe actions based on the value function. More formally, the agent takes the action as

$$a_t = \begin{cases} \pi(a_t \mid s_t) & \text{if safe}(s_t, a_t) \\ \arg\max_{a_i \in \text{safe}(s_t, a_i)} V(s_{t+1}^i) & \text{if not safe}(s_t, a_t) \end{cases} \tag{1}$$

This allows the agent to never take the unsafe action, even while we are training the policy or are using a pretrained policy. The agent is only made to act after the action predicted by the policy goes through the safety check. This "plug and play" usage of the imaginative module allows us to scale safety among different RL agents.

In summary, before the episode starts we create an imaginative module for the agent starting from the starting state $s_0$ by performing roll-outs of the transition model $s_{t+1} \sim \mathcal{P}(s_t, a_t)$ and creating a graph which covers all the possible state-action pairs. We define the baseline state $\mathcal{B}$ which is taken as human input. The baseline state can either represent a safe state or an unsafe state. We then either perform the normal operation using a pretrained policy, or learn a new policy by augmenting it with the imaginative module to ensure safety even while learning. The action $a_t$ predicted by the policy $\pi(a_t|s_t)$ is checked for safety using the imaginative module as described above. If the $a_t$ is safe, it is performed - otherwise, we act greedily among all the safe actions wrt. value function $V(s_{t+1}^i)$ where $s_{t+1}^i$ is the next state the agent reaches if it performs $i^{th}$ safe action.

## 5 EXPERIMENTS

To test the incorporation of the imaginative module as proposed in the paper, we consider environments in which no negative reward is provided to the agent when it is about to take an unsafe action. To adhere to the baselines and to compare it with existing formulations, we test our methods on the AI Safety Gridworld Environments (Leike et al. (2017)) related to avoiding negative side effects. To test the practicality of our method in a real-world-like environment, we train an agent to drive a car safely without getting off the track. The choice of baseline state $\mathcal{B}$ is specific to each environment and is made by considering which is easier to define, a set of safe states or a set of unsafe states.

### 5.1 EXPERIMENTAL SETTINGS

For the gridworld environments, imaginative module is modeled using two convolutional neural networks (CNNs) to estimate the transition function $s_{t+1} \sim \mathcal{P}(s_t, a_t)$ and the reward function $r_t = \mathcal{R}(s_t, a_t)$. Both of these networks share a common CNN based encoder which brings the current state $s_t$ to an embedding space. This embedding is passed through the separate networks of the transition and reward functions to predict the next state $s_{t+1}$ and the reward $r_t$. The reward and the state space are discretized and this is treated as a classification problem. Both the state loss and reward loss are cross-entropy losses over the softmax activation function.

The final loss is given as

$$\text{Total Loss} = \text{State Loss} + \alpha \times \text{Reward Loss}, \tag{2}$$

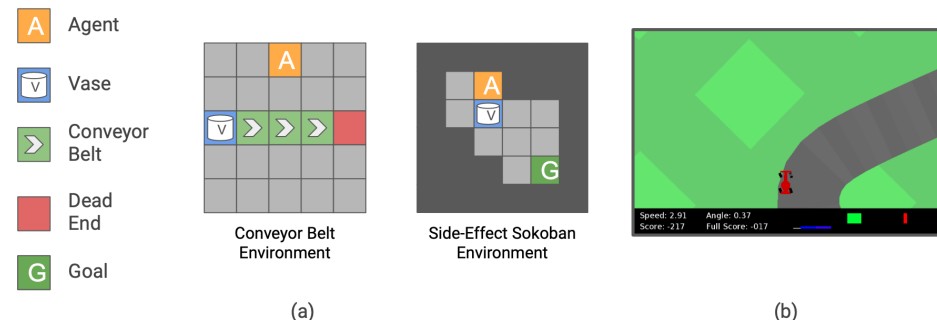

Figure 2: *(a)* The gridworld environments are specifically designed to test agents against negative side effects. Usage of imaginative module ensures that the agent does not take actions which lead to negative side effects in the future. *(b)* The car simulator environment used to train the agent to drive safely without going off the track. The baseline states here represents unsafe states.

where $\alpha$ is a regularization coefficient which provides the required weight to the reward loss. In our experiments, the value of $\alpha$ is kept as $0.1$. We use Adam Optimizer (Kingma & Ba (2014)) to minimize the loss.

To collect distinctive transition tuples $(s_t, a_t, s_{t+1}, r_t)$ for training the transition dynamics, we deploy multiple agents to run in parallel. All of these agents act randomly and explore different aspects of the environment. In our experiments, we used 16 agents running in parallel. The transition dynamics model is trained for 5000 total timesteps. The imaginative module over the gridworld environment reports a ROC AUC score of 0.95.

For the driving environment we model the imaginative module using rollouts from the actual environment. This is done to assure that the imaginative module doesn't deviate from the actual environment, since the transition dynamics in this environment are hard to learn due to their continuous state space.

The policies are trained using the standard on-policy A2C method. The agent is modeled using a CNN with a common encoder. The embeddings from the encoder are passed to 2 networks with 2 convolutional layers and 2 dense layers – representing the policy and the value function. The network is optimized using the RMSProp (Tieleman & Hinton (2012)) optimizer with learning rate as $7 \times 10^{-4}$. Whenever these policies predict an action, the action is checked for safety with the imaginative module. If the action is unsafe, it is avoided and the policy is made to act greedily based on the value function among all the safe actions.

## 5.2 GRIDWORLD ENVIRONMENTS

We test the incorporation of the imaginative module on environments focused on avoiding negative side effects from the AI Safety Gridworlds suite (Leike et al. (2017)). Specifically, we test our proposal on "Side Effect Sokoban" and "Conveyor Belt" environments to consider different categories of negative side effects that can occur in a dynamic environment. The state representation is a 5x5 matrix consisting of different numbers representing different sprites. At every time step, the agent can take actions (up, down, left, right). The interference behavior in both of the environments consists of moving an obstacle (by stepping into the square containing the obstacle).

The Side Effect Sokoban environment consists of a vase that needs to be pushed out of the way for the agent to reach the goal. The agent is provided a reward of -1 for every step it takes and a final reward of +50 if it reaches the goal. The unsafe behavior is taking the shortest path to the goal, which involves pushing the vase down into a corner (an irrecoverable position). No negative reward is provided to the agent when it does takes unsafe behavior. The desired behavior is to take a slightly longer path by push the vase to the right. The action of moving the vase is irreversible in both cases: if the vase is moved to the right, the agent can move it back, but then the agent ends up on the other side of the vase. Thus, the agent receives the maximum penalty of 1 for moving the vase in any direction, so the penalty does not incentives the agent to choose the safe path.

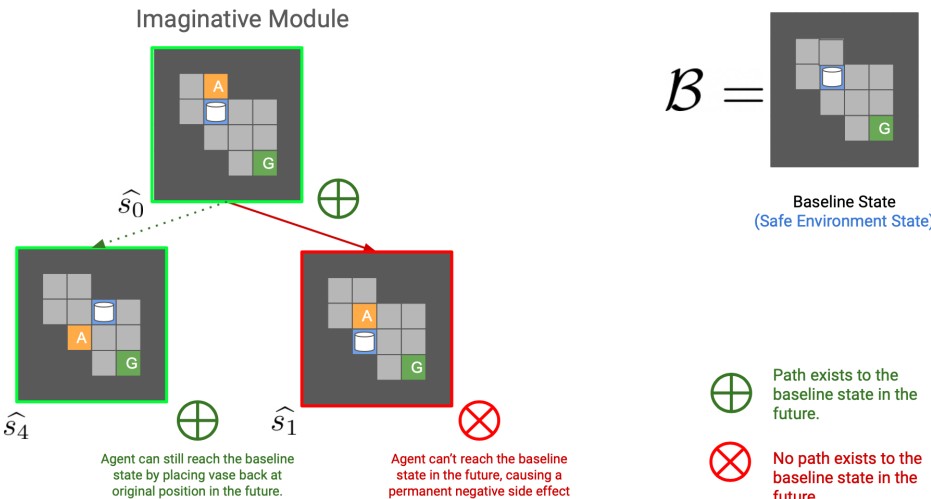

Figure 3: **Illustration of our approach on the Side Effect Sokoban environment.** The baseline state $\mathcal{B}$ is chosen as the environment state in which the obstacle is in its original position, hence, representing a safe state. Based on the imaginative module, a safe action is the one which allows the agent to reach the baseline state in the future.

We take the initial environment state (without the agent sprite) as the baseline state, and consider it as a safe state. An action $a_t$ is considered to be a "safe action" if there's at least one path from the next state (which it reaches after taking $a_t$) to the baseline state in the imaginative module. Hence, whenever the policy takes an action $a_t$, the imaginative module is used to check if the action is safe (i.e., if the agent can reach the initial environment state). Hence, the action which causes the vase to move into the corner is always avoided as if it was taken, it'd be impossible to reach the baseline state. This action is even avoided while a policy is being learned.

The Conveyor Belt environment consists of a conveyor belt that moves to the right by one square after every agent action. There is a vase on the conveyor belt which will break if it reaches the end of the belt. The agent receives a reward of +50 for taking the vase off the belt. The desired behavior is to move the vase off and then stay put. However, it is likely that the agent moves the vase off the conveyor belt and then back on as there's nothing avoiding it to do so.

In this case, we define the baseline state as the one in which the vase is broken after reaching the end of the belt - and it represents an unsafe state. An action $a_t$ is considered to be an "unsafe action" if all the paths from the next state $s_{t+1}$ lead to the specified baseline state in the imaginative module. Hence, the action which causes the vase to be moved back into the conveyor belt is always avoided. If the action had been taken, it would undoubtedly cause the vase to break.

### 5.3 AUTONOMOUS CAR ENVIRONMENT

To use the imaginative module for safe exploration, and to test the practicality of our method using environments based on real-world tasks, we incorporate it on a policy which learns to drive a car in a modified version of the Car Racing environment provided by OpenAI gym (Brockman et al. (2016)). The policy has to learn how to drive safely by never getting off the track. We discretize the existing continuous action space to be able to create the imaginative module. The environment is modified to so that no negative reward is provided to the agent when it gets off track. To ensure that the imaginative module doesn't deviate from the actual environment due to biases in the learned transition dynamics, we create the graph using rollouts from the actual environment.

We define the set of states in which the car is off-track as the baseline state space $\mathcal{B}$. An environment function tells whether the car is within the track or not. An action $a_t$ is considered to be a "unsafe action" if all the paths from the next state $s_{t+1}$ lead to the specified baseline state in the imaginative module. Due to the huge size of the state space, we create an imaginative module looking $k$-steps

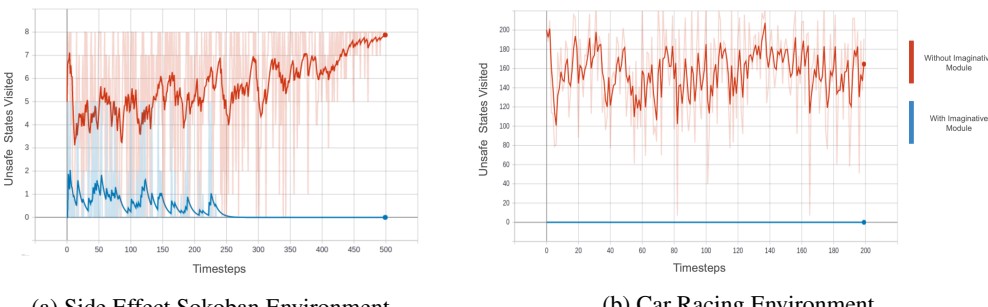

(a) Side Effect Sokoban Environment             (b) Car Racing Environment

Figure 4: **The Imaginative Module reduces visits to unsafe states.** We plot the number of unsafe states visited by the agent in the respective environments, shown across training steps. Without any safety constraints, the agent visits a high number of unsafe states. Incorporation of the imaginative module reduces the number of unsafe states visited even while training. The plot shows the average across 3 random seeds, with the dark line depicting an exponential moving average of the light line.

into the future, at every $k^{th}$ time step. Hence, the action is considered as unsafe if it causes the car to go off track in $l$-steps into the future, where $l < k$.

In Figure 4, we plot how frequently the agent visits unsafe states throughout training. As compared to the baseline in which the agent doesn't use an imagination module for planning, we observe that our method visits significantly fewer unsafe states. In the Side Effect Sokoban environment, the agent takes the unsafe and the shortest path if no implicit safety constraints are provided. This causes the agent to visit more unsafe states as the policy improves. In the Car Racing environment, we count the time steps in which the agent remains off track. The use of our approach causes the agent to never get off track, even if the policy is acting randomly.

## 6    CONCLUSION

This paper proposed the usage of a learned imaginative module, as a "plug-and-play" approach to promote safe exploration and avoid negative side effects by the agent. Our method allows any policy, either pretrained or a one being learned to act safely. We validate our approach by experimenting it on two gridworld environments and a car simulator environment. We do realize that incorporating this approach in real-world task with complex environments would require significant advancement and work in learning how to model the environment, but we believe this first step could provide some direction to more work to be done in this area.

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
