# OpenReview forum: "Avoiding Negative Side-Effects and Promoting Safe Exploration with Imaginative Planning"
_ICLR.cc/2020/Conference — Reject_

### Official Review · AnonReviewer1 · 2019-10-15
**Official Blind Review #1**

**Rating:** 1

**Review:**

Thie paper proposes using an "imagination" module to provide safe exploration during RL learning. The imagination module is used to perform forward predictions, constructing a graph between possible states. If any action would lead to a "base state" that is an unsafe state that action will not be executed and another "safe" action is selected from the policy.

I have a number of comments and questions after reading the paper:
- How do you get the forward model to be usably accurate? You do say that the model is a CNN model and is shared to learn the reward function as well. In the paper it says your method will lead to the agent never reaching an unsafe state, do you train the network in some way to make sure it does not make an inaccurate prediction around the unsafe state?
- There is a lot of repetitive content in the paper that can be discarded to condense down the paper and make it more readable.
- It would be nice to see more tasks or at least one that was more realistic... The tasks used in the paper appear to be common ones but they still feel rather artificial. Also, it seems in the paper there are only learning curves for 2 of the 3 tasks.
- In the related work you say "However, these methods are difficult to quantify and depend a lot on the diversity of settings in which they are performed." Can you expand on this? Why do they depend on these issues so much? The motivation for your method stems from these methods not being good enough, so further detail on this facet is important.
- If you are using environments with discrete actions and performing prediction I am not sure if that can be called imaginative. Rather it should be called sampling. The forward model does not even appear to be stochastic.
- The description of the imagination module training is not very clear. Is it trained on 5000 tuples or is the network trained for 5000 updates? There needs to be much more detail on this process.

Overall the method seems interesting but does not appear to be a significant improvement.

**Experience Assessment:**

I have published one or two papers in this area.

**Review Assessment: Checking Correctness Of Derivations And Theory:**

I carefully checked the derivations and theory.

**Review Assessment: Checking Correctness Of Experiments:**

I carefully checked the experiments.

**Review Assessment: Thoroughness In Paper Reading:**

I read the paper thoroughly.

---

### Official Review · AnonReviewer2 · 2019-10-22
**Official Blind Review #2**

**Rating:** 1

**Review:**

This paper presents a model-based approach to safety in RL, where the agent uses a transition model to plan ahead to avoid actions that can lead it to unsafe states. They call the planning component an imaginative module. The agent takes the baseline state as input - that can be used to define either a safe or unsafe state, that is used in the planning component. The authors claim that using these two techniques they can tackle both the safe exploration (not violating safety constraints during learning) and irreversible side-effects (unintended irreversible behavior due to poorly designed reward-function).  They validate their approach on two grid world environments and self-driving car simulators.

This paper should be rejected because of the assumptions it makes goes against the very task they are trying to solve. In the sense, the task is trivial given the assumptions they have.

1) The inconsistent assumption regarding the access to trajectories to learn a model.
The authors start with the assumption that the agent does not have access to the model (Sec 3) , and they explicitly learn the model. However, in the very next section (Sec 4), the authors assume that they can deploy a number of agents that interact with the environment randomly and collect that data to learn a complete transition model. Note that this assumption is wrong because:
If the random data agents are “safe”, i.e., don’t violate any safety constraint or cause any harmful behavior in the environment, then it is equivalent to assuming the agent having access to all the data to learn the model. This is a very big assumption that essentially says the agent has access to the model, which defeats the purpose of the safe-exploration problem.
If the random agents are “unsafe”, i.e., they can violate the safety constraint, then it goes against the very claim made about their method being able to respect the constraints throughout the learning process.

2) The assumption about the baseline state(s).
This is also a pretty big assumption to have, that is not acknowledged in the paper. If the agent already has the set of all the states it needs to avoid (or the set of states that are safe), then along with the assumption regarding access to the model, solving reversibility is significantly easier task then the general safe exploration problem [1, 2]

3) The results reported in Figure 4 are not statistically significant. The experiments are only run over 3 random seeds [3]

4) Can you give a few more details about the assumptions? In terms of how realistic they are or how essential they are to the method.


Things to improve the paper that did not impact the score:
- The negative side-effects problem that this work address is only based on reversibility criteria.
Claim about learning the dynamics model is sample efficient is unsupported.



References:
[1] Berkenkamp, Felix, et al. "Safe model-based reinforcement learning with stability guarantees." Advances in neural information processing systems. 2017.

[2] Dalal, Gal, et al. "Safe exploration in continuous action spaces." arXiv preprint arXiv:1801.08757 (2018).

[3] Henderson, Peter, et al. "Deep reinforcement learning that matters." Thirty-Second AAAI Conference on Artificial Intelligence. 2018.


**Experience Assessment:**

I have published one or two papers in this area.

**Review Assessment: Checking Correctness Of Derivations And Theory:**

N/A

**Review Assessment: Checking Correctness Of Experiments:**

I assessed the sensibility of the experiments.

**Review Assessment: Thoroughness In Paper Reading:**

I read the paper thoroughly.

---

### Official Review · AnonReviewer3 · 2019-10-23
**Official Blind Review #3**

**Rating:** 1

**Review:**

This paper proposes to use learned transition models to do two separate things: (i) avoid unsafe states and (ii) allow an alternative channel for task reward specification. The idea is to create a comprehensive connectivity graph of the states in the environment. Once done, an agent can avoid unsafe states by avoiding states that are unconnected to a specified safe state. A practitioner might also specify safe/unsafe states as an additional source of information about the reward.

This paper suffers from poor and loose writing, incomplete specification of its experiments, unrealistic assumptions during evaluation (Sec 5.3 "we create the graph using rollouts from the actual environment" to avoid errors from learning a transition model).

The paper does not address basic concerns with its approach: how is the model to be learned at all, if it is to be comprehensive in the way that is necessary for the connectivity graph (which this paper calls an "imaginative module")? The authors say this is done through multiple agents performing random actions in the environment, in which case, isn't this extremely unsafe training time by the paper's own definition of safe exploration?

Further, creating a complete connectivity graph is unrealistic even for fully known transition models in most reasonably complex settings, such as, say, Go or Chess.

If the transition model is fully known as in the car racing setting, why not directly use that to plan and solve the game?

Experiments show fewer "unsafe" states for the paper's approach compared to a method that has no way to know that those states are unsafe. How is this a reasonable validation, especially when the transition model is fully known? Also, this is an insufficient metric by itself as it says nothing about whether the method actually performed well at the task.

**Experience Assessment:**

I have read many papers in this area.

**Review Assessment: Checking Correctness Of Derivations And Theory:**

I did not assess the derivations or theory.

**Review Assessment: Checking Correctness Of Experiments:**

N/A

**Review Assessment: Thoroughness In Paper Reading:**

I read the paper at least twice and used my best judgement in assessing the paper.

---

### Decision · Program_Chairs · 2019-12-19

**Decision:**

Reject

**Comment:**

This paper tackles the problem of safe exploration in RL. The proposed approach uses an imaginative module to construct a connectivity graph between all states using forward predictions. The idea then consists in using this graph to plan a trajectory which avoids states labelled as "unsafe".

Several concerns were raised and the authors did not provide any rebuttal. A major point is that the assumption that the approach has access to what are unsafe states, which is either unreasonable in practice or makes the problem much simpler. Another major point is the uniform data collection about every state-action pairs. This can be really unsafe and defeats the purpose of safe exploration following this phase. These questions may be due to a miscomprehension, indicating that the paper should be clarified, as demanded by reviewers. Finally, the experiments would benefit from additional details in order to be correctly understood.

All reviewers agree that this paper should be rejected. Hence, I recommend reject.